# Wireless Single-Lead versus Standard 12-Lead ECG, for ST-Segment Deviation during Adenosine Cardiac Stress Scintigraphy

**DOI:** 10.3390/s23062962

**Published:** 2023-03-09

**Authors:** Luna Fabricius Ekenberg, Dan Eik Høfsten, Søren M. Rasmussen, Jesper Mølgaard, Philip Hasbak, Helge B. D. Sørensen, Christian S. Meyhoff, Eske K. Aasvang

**Affiliations:** 1Department of Anesthesiology, Centre for Cancer and Organ Diseases, Rigshospitalet Copenhagen University Hospital, Blegdamsvej 9, 2200 Copenhagen, Denmark; 2Department of Cardiology, Rigshospitalet Copenhagen University Hospital, 2100 Copenhagen, Denmark; 3Biomedical Signal Processing & AI Research Group, Digital Health Section, Department of Health Technology, Technical University of Denmark, 2800 Kgs. Lyngby, Denmark; 4Department of Clinical Physiological and Nuclear Medicine, Center for Diagnostics, Rigshospitalet Copenhagen University Hospital, 2100 Copenhagen, Denmark; 5Department of Anaesthesia and Intensive Care, Copenhagen University Hospital-Bispebjerg and Frederiksberg Hospital, 2400 Copenhagen, Denmark; 6Department of Clinical Medicine, University of Copenhagen, 2200 Copenhagen, Denmark

**Keywords:** single-lead, ECG, wireless, reversible ischemia, validation, ST segment

## Abstract

Wearable wireless electrocardiographic (ECG) monitoring is well-proven for arrythmia detection, but ischemia detection accuracy is not well-described. We aimed to assess the agreement of ST-segment deviation from single- versus 12-lead ECG and their accuracy for the detection of reversible ischemia. Bias and limits of agreement (LoA) were calculated between maximum deviations in ST segments from single- and 12-lead ECG during ^82^Rb PET-myocardial cardiac stress scintigraphy. Sensitivity and specificity for reversible anterior-lateral myocardial ischemia detection were assessed for both ECG methods, using perfusion imaging results as a reference. Out of 110 patients included, 93 were analyzed. The maximum difference between single- and 12-lead ECG was seen in II (−0.019 mV). The widest LoA was seen in V5, with an upper LoA of 0.145 mV (0.118 to 0.172) and a lower LoA of −0.155 mV (−0.182 to −0.128). Ischemia was seen in 24 patients. Single-lead and 12-lead ECG both had poor accuracy for the detection of reversible anterolateral ischemia during the test: single-lead ECG had a sensitivity of 8.3% (1.0–27.0%) and specificity of 89.9% (80.2–95.8%), and 12-lead ECG a sensitivity of 12.5% (3.0–34.4%) and a specificity of 91.3% (82.0–96.7%). In conclusion, agreement was within predefined acceptable criteria for ST deviations, and both methods had high specificity but poor sensitivity for the detection of anterolateral reversible ischemia. Additional studies must confirm these results and their clinical relevance, especially in the light of the poor sensitivity for detecting reversible anterolateral cardiac ischemia.

## 1. Introduction

Hospitalized patients are at risk for severe complications occurring in 30% of major surgical patients within the first 30 days [1,2], and they often go undetected with the subsequent risk of progression and permanent injury. A frequent and severe complication is cardiac injury, which is common in both medical and surgical patients and is significantly related to increased morbidity and mortality [3,4,5,6,7].

While many cases of out-of-hospital myocardial infarctions are accompanied by characteristic symptomatology such as dyspnea and chest pain, most myocardial injuries in the postoperative phase remain silent and with minimal or no discomfort, in part due to concomitant surgical wound pain and analgesic administration [8,9]. This subsequently may result in the delay or even absence of prevention and treatment, with increased risk for heart failure [10] and mortality being 35% vs. 45% for patients with or without recognized myocardial infarction, respectively [11].

Despite the described risk for cardiac injury there is no routine monitoring of noncardiac-hospitalized patients, except for specific patient cases with known severe preoperative ischemic coronary disease, subjective complaints or where intra- and Post Anesthesia Care Unit (PACU) monitoring has detected ischemic ECG-signs, where a 12-lead ECG is recommended. However, a wired portable 12-lead ECG, although non-invasive, puts restraints on the patient’s movements and is, for all practical purposes, therefore not possible to wear all the time, especially in the optimized hospital setting where early mobilization is critical to enhance recovery and avoid adverse events related to inactivity [12]. Furthermore, the future hospital systems will see a constant drive towards early discharge to the patient’s own home, calling for true wearable wireless monitoring devices. Wireless continuous ECG monitoring with artificial intelligence real-time analysis for staff alerts could potentially enable large-scale patient monitoring without hindering activity. However, before such systems are implemented, agreement with clinical standard needs to be assessed, including the accuracy for ischemia detection.

We aimed to assess the agreement of a single-lead wireless ECG versus a standard 12-lead wired ECG regarding detection of ST deviations during an adenosine stress test, in patients suspected for myocardial ischemia. The second aim was to describe the diagnostic accuracy of both methods (single- and 12-lead ECG) to diagnose reversible antero-lateral myocardial ischemia during Positron Emission Tomography (PET) myocardial perfusion imaging [13]. The decision to only evaluate antero-lateral ischemia on PET myocardial perfusion imaging is based on the physical shape of the equipment allowing it to best be placed in a formation theoretically correlating to this area of ischemia.

## 2. Methods

### 2.1. Study Design

This was an exploratory study performed at the Department of Clinical Physiological and Nuclear Medicine, Center for Diagnostics, Rigshospitalet, Copenhagen, Denmark from 3rd January 2020 until 30th July 2020. 

### 2.2. Patients

Patients referred for a ^82^Rubidium PET Computer Tomography myocardial perfusion imaging (cardiac-PET-CT) scan due to suspicion of myocardial ischemia and above 18 years were included consecutively. Preoperative assessments were included as well.

Patients without ECG recordings, or where the two ECGs could not be time-matched due to technical problems, were excluded.

No formal power analysis was performed due to lack of knowledge on the viability between the two ECG methods, but aimed at including ≥100 patients.

### 2.3. Data Collection

ECG data were collected before and during an adenosine infusion at 140 μg/kg/min for 6 min with the patient resting in supine position. ECG recordings were obtained from a single-lead ECG patch (Lifetouch, Isansys, Oxfordshire, UK), placed with one electrode in the fourth intercostal space at the left sternal border and the other in the left anterior/midaxillary line, theoretically monitoring the anterolateral myocardium (Figure 1a) [14,15]. Simultaneous recordings were obtained from a 12-lead ECG (ST-104 12-lead ECG, Schiller AG Baar, Schmeiz) with standard electrode placements. Data from the single-lead ECG were transmitted via Bluetooth to a gateway and onwards to a local database server via the secure hospital wi-fi. Data from the 12-lead ECG were collected from the receiving computer.

The 82Rb PET/CT myocardial perfusion scans were performed using a hybrid PET/CT scan in 3D mode (Siemens Biograph mCT 128, Siemens, Munich, Germany) after administration of 1100 MBq 82Rb (CardioGen-82, Bracco Diagnostics, Milan, Italy) via an intravenous canula placed in the lower arm. (Figure 1b).

### 2.4. Outcomes

The primary outcome was maximum ST-segment deviations from the baseline in the single-lead ECG and simultaneous 12-lead ECG (lead II, V5 and V6) or during the adenosine stress test. We defined mean_diff_ as clinically acceptable if within ±0.02 mV.

A secondary outcome was the frequency of patients with ST deviation ≥ ±0.1 mV on 12-lead ECG compared to ST-deviation ≥ ±0.1 mV on single-lead ECG [16]

Exploratory outcomes were the sensitivity and specificity of ST-deviation detection ≥ ±0.1 mV on 12-lead ECG and single-lead ECG against reversible anterolateral cardiac ischemia assessed by cardiac perfusion (cardiac-PET-CT).

### 2.5. Variables

Demographic data (age, height, weight, gender, comorbidity, and medications) were collected from the patients’ medical journal on the day of the investigation (Table 1).

### 2.6. ECG Data Analyses

The two ECG sampling methods were synchronized 1:1 by timestamps and heart rate variations.

The automatic estimation of ST-deviations was based on the location of the R- and T-wave tops in each heart cycle. The reference level was estimated as the voltage level at the point with the lowest change in signal in an area of 200–50 milliseconds prior to the R-peak location. The ST-level was estimated as the voltage level at the point with the lowest change in signal in the area between the R-peak and T-wave top. The ST deviation was computed as the difference between the reference and ST level, and the mean value in a segment of 10 s was used for the analysis. To validate the method, a software tool was developed using Matlab 2019a (undisclosed), in which it was possible for a physician to manually mark a location of the reference and ST level, and by automatic estimation, get an average in a 10-s segment. The manual annotation was used to validate the automatic method for ST-level estimation, and the validation was done on 300 10-s segments randomly chosen from the first 40 patients included in the study. The validation showed a mean absolute error of 0.0043 mV between the automatic estimate and manual annotation of the ST-level.

The single-lead ECG and 12-lead ECG were interpolated to create datapoints for every second. Following this, both ECGs were corrected for baseline using a part of the ECG obtained during the dynamic PET rest scan. A rolling mean of 60 s for these datapoints was used for the primary analysis, by taking the average of the first 60 s of recording, followed by a frame shift forward for every one second, including the next number and excluding the first number in the frame (Figure 2).

The rolling means of both ECG methods in the same time frame were used to calculate the maximum delta value as the stress-test period minus rest period and used for Bland Altman plots and LoA statistics.

## 3. Statistics

Baseline data are presented by descriptive statistics including means with 95% confidence intervals (95% CI), or medians with range and interquartile ranges (IQR), or numbers and percentages where appropriate.

The primary analysis was the agreement of the single-lead ECG versus the 12-lead ECG. Bland Altman plots were used to illustrate mean differences (mean_diff_) and limits of agreement (LoA, 1.96 mean-difference x SD) with 95% CI for the LoA.

The secondary and exploratory analysis included diagnostic accuracy assessment by sensitivity and specificity for the detection of reversible anterior and/or lateral myocardial ischemia, for both ECG methods, defined as V5, V6, and II. Myocardial ischemia was based upon experienced clinical assessments of the ^82^Rb PET/CT myocardial perfusion scans.

Statistical analysis was performed using Python (Version 3.7.6.; Python Software Foundation, Beaverton, OR, USA), R (version 3.6.1.; R Foundation, Vienna, Austria) and Matlab (version 2019a; MathWorks, MA, USA).

## 4. Results

A total of 110 patients were included during the study period and 17 were excluded, 14 because of poor quality of the signal leading to dissatisfactory matching between ECG methods and three due to interrupted scan sequences, leaving 93 patients for analysis. Mean age was 63 years (SD 10.7) with an average BMI of 29, and30.07% were women (31 patients). The most frequent co-morbidities were diabetes in 58 patients, hypertension in 22, and 20 were diagnosed with ischemic cardiac disease before the scan.

Totals of 26 h of 12-lead ECG and 31 h of single-lead ECG were recorded counting all patients included, with an average of 16.2 min of continuous recording per patient on the 12-lead ECG and 20.2 min of continuous recording per patient on the single-lead ECG.

After matching of the recordings on heart rate, interpolation, and exclusion of data according to exclusion criteria, we had a total of 93 datapoints corresponding to the maximum value of the rolling mean for each patient.

### 4.1. Primary Outcome

The agreement between the 12-lead and single-lead ECG is shown in the Bland Altman plots in Figure 3A–C. Summarizing, mean_diff_ lead II vs. single lead −0.019 mV (95% CI: −0.030 to −0.009), mean_diff_ lead V5 vs. single lead −0.005 mV (95% CI −0.021 to 0.010) and mean_diff_ V6 vs. single lead −0.006 mV (95% CI −0.021 to 0.010). The widest LoA was seen in lead V5 vs. single lead; upper LoA 0.145 mV (95% CI:0.118 to 0.172) and lower LoA −0.155 mV (95% CI: −0.182 to −0.128).

### 4.2. Secondary Outcomes

On the 12-lead ECG, significant ST deviations (≥0.1 mV) in either II, V5, and/or V6 were observed in nine patients (9.7%). Seven patients had significant ST-deviations in II (7.5%), five in V5 (5.4%) and two in V6 (2.2%) (Table 2). On the single-lead ECG, significant ST-deviations were observed in nine patients (9.7%). Four patients with ≥0.1 ST deviations in the single-lead ECG also had ≥0.1 mV ST deviations in the 12-lead ECG (44.4% overlap) (Table 3).

### 4.3. Exploratory Outcome

Twenty-four patients had reversible lateral or anterior ischemia on their cardiac-PET-CT, two of which had significant ST-segment deviations on the single-lead ECG, resulting in a sensitivity of 8.3% (CI: 1.0%; 27.0%) and a specificity of 89.9% (CI: 80.2%; 95.8%) (Table 3). In the 12-lead ECG, 3 of the 24 patients had significant ST deviations, resulting in a sensitivity of 12.5% (CI: 3.0%; 32.4%) and a specificity of 91.3% (CI: 82.0%; 96.7%) 

Comparison of the algorithm developed to detect ST-segment deviations in recordings from single-lead ECG and annotations made by a physician showed a mean of 0.004 mV with an upper LoA of 0.079 mV and lower LoA of −0.071 mV.

**Table 3 sensors-23-02962-t003:** 2 × 2 table showing the number of patients with significant ST deviations ≥ 0.1 mV on the single-lead ECG and how they relate to the results from the cardiac-PET divided into reversible anterolateral ischemia or no reversible anterolateral ischemia.

	Reversible Lateral/Anterior Ischemia	No Reversible Lateral/Anterior Ischemia	Total
Single-lead ST-deviations ≥ 0.1 mV	2	7	9
No single-lead ST-deviations ≥ 0.1 mV	22	62	84
Totals	24	69	93

## 5. Discussion

This is, to our knowledge, the first clinical trial evaluating the accuracy of a single-lead ECG in detecting clinically relevant ST-segment changes during a physiological stress test in patients with suspected coronary artery disease. It is also the first study to correlate findings on a single-lead ECG with documented ischemia using cardiac PET, a diagnostic modality known to have a very high sensitivity for detecting myocardial ischemia. The ability of continuous ST-segment monitoring to detect both symptomatic and asymptomatic ischemic episodes is well documented, but currently relies on multi-lead monitors, whereas the use of single-lead ECG patches has mainly been evaluated in the scope of arrhythmia detection. There are, however, data supporting the potential of single-lead ECG in morphology detection. In a study by Rajbhandary et al., morphological representation and reasonable timing accuracy of ECG signals from a patch sensor compared to 12-lead ECG was documented [17]. However, this study emphasized overall ECG morphology and was not designed to address analysis of ST-segment variation over time. Jennings et al. studied ST-segment deviations in 44 patients undergoing controlled coronary occlusion using a 120-lead chest map, and analyses identified chest positions that theoretically could be suitable for the application of an ECG patch, but relying on a combination of at least four leads [18]. Furthermore, their study demonstrated that the diagnostic ability varied according to which coronary was affected. In a study by Gibson et al., a large database of 12-lead ECGs from patients with and without ST-segment elevation myocardial infarctions was used to investigate the potential of single-lead ECG detection of myocardial ischemia [19]. The authors found reasonable sensitivity and specificity, but like in the study by Jennings et al., performance differed depending on which coronary artery was affected. Furthermore, patients with ST-segment elevation myocardial infarctions usually have substantial and transmural myocardial ischemia, and it is therefore unclear whether similar results apply when monitoring for less-severe degrees of clinically relevant myocardial ischemia. In a study by El-Hamad et al., QT variability measured using a single-lead ECG correlated reasonably with 12-lead ECG and differed in patients with and without ischemia [20]. However, the clinical relevance of QT variability in ischemia detection remains uncertain and is not recommended in current guidelines [21].

The variations, represented by LoAs, were between −0.155 and −0.145 mV; thus, there was a much larger variation within the cohort and outside the prespecified clinically acceptable 0.02 mV. We found a maximum mean_diff_ in the ST segment during an adenosine stress test of −0.019 mV between a standard 12-lead and a wireless single-lead ECG. However, when considering the definition of diagnostic ST depression made by European Society of Cardiology as ST deviation ≥ 0.1 mV observed 60–80 milliseconds after the J-point, in one or more ECG leads [10], the observed LoAs are likely to result in the clinical situation where significant ischemia will be detected on both single- and 12-lead ECG. However, further studies correlating the myocardial injury assessed by troponins versus the ST changes are warranted.

The simple ST-threshold analysis had poor sensitivity for reversible ischemia for both the 12-lead and the single-lead ECG when compared to the gold-standard cardiac-PET-CT (12.5% and 8.3%), similar to previous studies [13], requesting for more advanced analysis of ECG and the advantages of the wearable technology in order to gain its full potential. Recent deep neuronal network analysis suggests a good accuracy, potentially allowing for continuous monitoring and early interventions if the finding can be confirmed in prospective studies using wearable technologies, where the bias found in our study must be incorporated. When comparing results from the cardiac-PET-CT, the single-lead and the 12-lead ECG identified 2/24 and 3/24 patients, respectively, with reversible lateral and/or anterior ischemia. This is in line with research showing that cardiac-PET-CT has a much higher sensitivity in diagnosing obstructive coronary disease than 12-lead ECG exercise testing (90%) [22]. Our study also shows that in cases with significant (>0.1 mV) elevation on either single or 12-lead ECG, there were 44% overlap. Combined with the cardiac-PET results, the findings show that single- and 12-lead ECG ST changes alone cannot yet be used for reversible ischemia detection, but based upon the acceptable agreement, can be used as a clinical support tool for instigating further investigations such as clinical examination and biomarkers [23,24]. However, ST-segment analysis alone will not suffice, and future studies should investigate automated assessment of morphological changes of the total ECG segments for diagnostic accuracy of ischemia [19].

A main strength of the current study is the stringent methodology, including that ECGs were all calibrated for a 2-min baseline ECG which is located before the administration of adenosine, reducing noise and artefacts. In addition, false-positive results on ST-depression analysis are seen to be more frequent in patients with abnormal resting ECGs [13], again reduced by baseline calibration.

The variation seen in the current study is also explained by the fact that we did not assess agreement during rest but decided to record during stress tests to allow for assessment of changes in ST segments similar to the intended use in a track-and-trigger clinical support system. In addition, we used clinically agreed thresholds for assessing sensitivity and specificity for ischemia detection, revealing that although acceptable agreement was found, this translates to high clinical specificity for both methods, although there was poor diagnostic sensitivity. The included patient population had various pre-risk profiles, again resembling the clinical population for which the system is intended. Finally, assessors were independently blinded to the study results, thus reducing bias.

The study comes with limitations, especially that the primary outcome did not consider whether the ST deviation that was used in the analysis was considered clinically significant. This was chosen as a secondary analysis, as the study aimed to assess the agreement, not only in patients with clinically significant deviation, but a mixed cohort resembling the intended clinical population. A larger cohort may also have resulted in less variation; however, it would not have improved the very low sensitivity observed from both ECG methods, again suggesting that continuous monitoring may be used for screening, but not diagnostics. We also pooled data from men and women, again resembling how the ECG patch would be used in the clinical setting. Another limitation to this study is that ischemia on PET myocardial perfusion imaging located outside the anterio-lateral region was not evaluated against the ST-changes. It is very possible that a study including other areas would provide additional information.

A 60-s rolling mean (also known as moving average) was used for the analysis of the single-lead and 12-lead ECG ST deviations. This method reduces the risk of noise and movements disturbing the results but also masks very short-lasting ST deviations above our set threshold that do not last long enough to get the rolling mean above the threshold. However, these are expected to be of minor clinical importance. Finally, the single-lead ECG is a single-component device with two ECG electrodes and cannot cover the same myocardial area as a 12-lead ECG, but this was the very purpose of the study, to see how well a single-lead would perform.

Obtaining a 12-lead ECG manually is time-consuming and in clinical practice is only performed when suspecting cardiac disease. Telemetry is not used routinely in the hospitalized setting beyond patients admitted to the cardiological ward due to limitations in mobilization and work to overview the telemetry readings. It is therefore of significant importance that continuous wireless monitoring of cardiac ischemia can be accomplished. Few studies have assessed deviating vital signs in the days and hours before postoperative myocardial injury is diagnosed, and emerging data suggest that desaturation, tachycardia, and tachypnoea are associated with upcoming myocardial injury [25]. This suggests that continuous wireless monitoring of traditional vital signs may provide earlier recognition of myocardial injury, and the ST-segment findings from our study can be directly incorporated for enhanced ischemia detection. With correction for the LoA observed, single-lead ST segment monitoring would still potentially detect ischemia cases that today go undetected whilst reducing the number of false alarms. We found a high specificity with both methods, which can be used to exclude critical cardiac ischemia. Incorporating the findings from this study in wireless warning systems assessing multiple vital signs will be a major step forward in increasing the postoperative cardiac monitoring. Based upon the presented findings, we suggest the next steps should include morphological ECG analyses rather than sole ST segments and investigating the importance of duration of ST deviations to potentially increase the diagnostic accuracy.

## 6. Conclusions

Agreement within predefined acceptable criteria for mean_diff_ was found between a standard 12-lead and single-lead ECG in detecting ST deviations during an adenosine stress test, suggesting a potential for clinical monitoring of ST-deviations, but with larger individual variation. The lowest mean_diff_ was found when analyzing single-lead versus the V5 lead of the 12 lead-ECG, but again with wider LoA. Both the single lead and 12-lead ECG had high specificity but poor sensitivity for detection of anterolateral reversible ischemia. Additional studies must confirm these results and their clinical relevance, especially in light of the poor sensitivity for detecting reversible anterolateral cardiac ischemia.

## Figures and Tables

**Figure 1 sensors-23-02962-f001:**
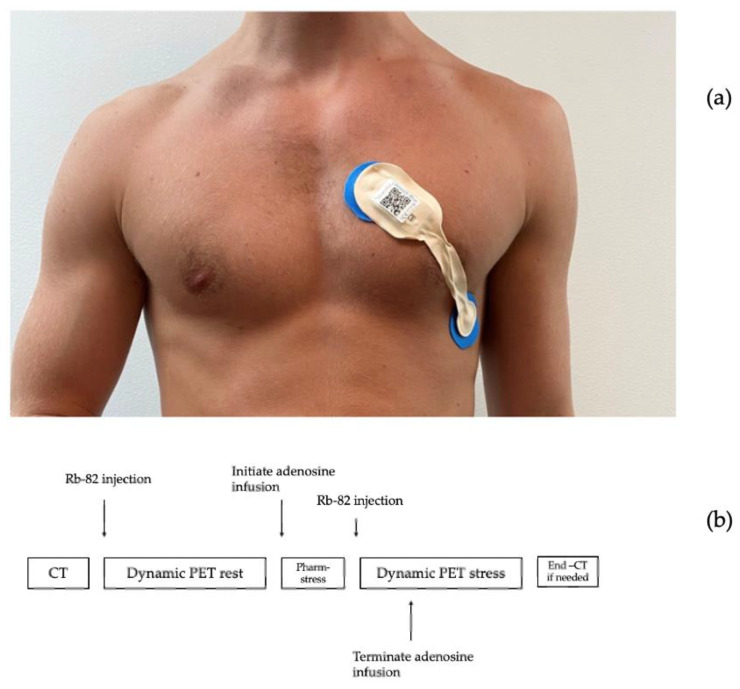
Graphic illustration of the cardiac-PET scan sequence (**a**): Placement of single-lead ECG patch (**b**): ^82^Rubidium PET Computer Tomography myocardial perfusion imaging sequence.

**Figure 2 sensors-23-02962-f002:**
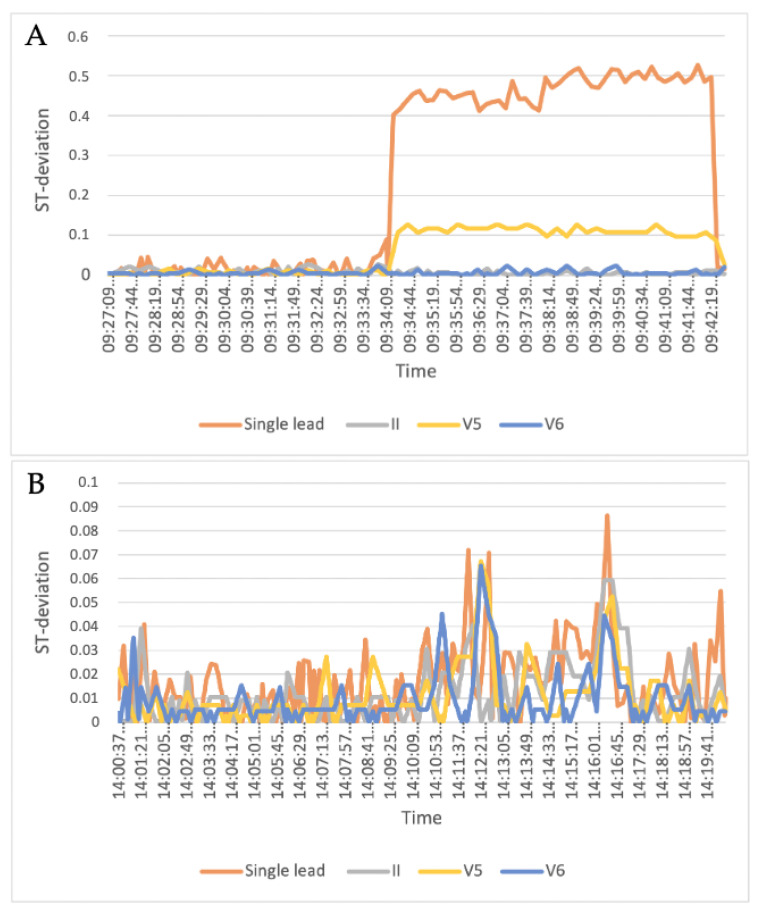
Examples of two patients’ ST-segment deviations during the adenosine cardiac stress test showing 12-lead II, V5 and V6 leads as well as the single-lead leads. (**A**) shows ST deviation > 0.1 mV in single lead and V5, whereas (**B**) does not show ST deviations > 0.1 mV in any recordings.

**Figure 3 sensors-23-02962-f003:**
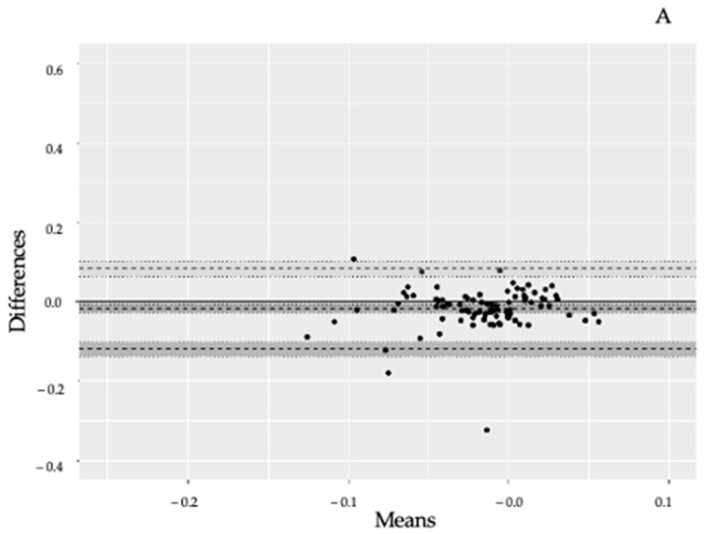
(**A**): Bland–Altman plot of II-lead and single-lead ECG results. Comparison of lead II from a 12-lead ECG versus a single-lead ECG on the anterolateral left chest. Limit of agreement = LoA Upper dotted line = upper LoA, lower dotted line = lower LoA. Dotted line in the middle = mean difference. Dark area around dotted lines = 95% confidence intervals (95% CI). Mean difference = −0.019 mV (95% CI: −0.030 to −0.009). Upper LoA = 0.082 mV (95% CI 0.064 to 0.100) and lower LoA = −0.121 mV (95% CI: −0.139 to −0.103). (**B**): Bland–Altman plot of V5-lead and single-lead ECG results. Comparison of lead V5 from a 12-lead ECG versus a single-lead ECG on the anterolateral left chest. Dotted line in the middle = mean difference. Limit of agreement = LoA. Upper dotted line = upper LoA, lower dotted line = lower LoA. Dark area around dotted lines = 95% confidence intervals (95% CI). Mean difference −0.005 mV (95% CI −0.021 to 0.010). Upper LoA 0.145 mV (95% CI: 0.118 to 0.172) and lower LoA −0.155 mV (95% CI: −0.182 to −0.128). (**C**): Bland–Altman plot of V6-lead and single-lead ECG results. Comparison of lead V6 from a 12-lead ECG versus a single-lead ECG on the anterolateral left chest. Limit of agreement = LoA Upper dotted line = upper LoA, lower dotted line = lower LoA. Dotted line in the middle = mean difference. Dark area around dotted lines = 95% confidence intervals (95% CI). Mean diff −0.006 mV (95% CI −0.021, 0.010). Upper LoA 0.138 (95% CI: 0.112; 0.164) and lower LoA −0.149 (95% CI −0.175; −0.123).

**Table 1 sensors-23-02962-t001:** Demographic data on patients having PET-CT with adenosine stress test. Age is given in years, weight in kilograms, height in centimeters, BMI = body mass index and is given in kg/m^2^. Ischemic Heart disease: chronic myocardial ischemia (17), stable angina (6), unstable angina (2)).

Patient Characteristics	Data Cohort n = 93
**Age**	63.1 (SD 10.7)
**Sex**	
Female	31
Male	62
**Height**	173 (SD 9.3)
**Weight**	87 (SD 17.8)
**BMI**	
Average (max, min)	29 (45; 13)
>30 (%)	58 (62)
<30 (%)	35 (38)
**Comorbilities**	
Diabetes (%)	38 (41)
Hypertension (%)	22 (24)
Ischemic cardiac disease (%)	20 (22)
Myocardial infarction (%)	10 (11)
Renal disease (%)	15 (16)
PCI/stent (%)	4 (4.3)
Heart failure (%)	7 (7.5)
Heart transplant (%)	5 (5.4)
Stroke (%)	3 (3.2)
Cardiomyopathy (%)	2 (2.2)
Pacemaker (%)	3 (3.2)
Peripheral vascular disease (%)	1 (1.1)
**Cause of referral**	
Other clinical trials (%)	31 (33.3)
Solid organ transplant (%)	13 (14.0)
Clinical evaluation (%)	49 (52.7)

**Table 2 sensors-23-02962-t002:** 2 × 2 table showing the number of patients with significant ST deviations ≥ 0.1 mV on the 12-lead ECG and how they relate to results from the cardiac-PET divided into reversible anterolateral ischemia or no reversible anterolateral ischemia.

	Reversible Lateral/Anterior Ischemia	No Reversible Lateral/Anterior Ischemia	Total
ST-deviations ≥ 0.1 mV on V5, V6 and/or II	3	6	9
No ST-deviations ≥ 0.1 mV on V5, V6 and/or II	21	63	84
Totals	24	69	93

## Data Availability

Data are potentially available upon request and pending internal review of such request.

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
