# Peer review of "Wireless Single-Lead versus Standard 12-Lead ECG, for ST-Segment Deviation during Adenosine Cardiac Stress Scintigraphy"

_sensors, 2023, doi:10.3390/s23062962_

Round 1

Reviewer 1 Report

This paper assessed the agreement of a single-lead wireless ECG versus a standard 12-lead wired ECG in judging ST-deviations for cardiac ischemia detection. Both methods had high specificity but poor sensitivity for detection of ischemia. The manuscript is well written and easy to follow. I have several minor comments.

1. Is the ST-deviation measurement algorithm reliable? The authors mentioned manual annotations were used for comparison but did not show this result in the text. 

2. Some figure examples of single-lead and 12-lead ECGs together with the analysis results will help readers better understand the paper.

3. The figures have low resolutions and should be improved.

Author Response

Reviewer 1:

Comment 1: Is the ST-deviation measurement algorithm reliable? The authors mentioned manual annotations were used for comparison but did not show this result in the text.

Reply: Thank you for the relevant comment and we agree that this information is important. The 12-lead ECG ST-segment analysis is CE-marked and as such should fulfil reliability requirements, although we cannot give details. As stated, we used a proprietary algorithm to extract ST-segment deviations from the ECG’s and compared it to clinical evaluation which showed almost perfect agreement but is reserved for a separate publication. We have added some more information to the segment on ECG data analysis, line number 136-142.

Comment 2: Some figure examples of single-lead and 12-lead ECGs together with the analysis results will help readers better understand the paper.

Reply: Thank you for the comment. We agree but are unfortunately not able to provide images of the 12-lead ECG as the recording is solely ST-segments and heartrate and does not include the graphical representation of the ECG. The graphical ECG is available for the single lead bit without the 12 lead is does not make sense. Therefore we have constructed two examples of ST-deviations showing the development of the single lead and 12 lead ECG during the cardiac stress test, in a patient without significant changes and a patient with significant changes, as Figure 2A and B, line number 150 to help the readers better understand the paper.

Comment 3: The figures have low resolutions and should be improved.

We have improved the resolution of the figures in the revised manuscript.

Reviewer 2 Report

The given manuscript work on - Wireless single-lead versus standard 12-lead ECG, for ST-segment deviation during adenosine cardiac stress scintigraphy, is quite interesting However; I found a few discrepancies that require to be tackled before publication.

1) The experimental results presented in Fig. 2 are quite confusing; moreover, the figures are of low quality, kindly improve the quality of the figure and text font size so that the audience may understand the data presented.

2) why did the authors have chosen traditional statistical analysis methods such as mean and standard deviations have been considering for analysis instead of using modern machine learning, regression, and classification methods?

3) The conclusion section missed the final analysis and factual/tabular conclusion statements about the overall analysis.

4) The authors have not explored the related works in the literature with in the manuscript. We suggest reading related works and logically presenting them in the manuscript in chronological order.

5) What is the actual author's contribution to the work? The author has not even clearly described anywhere their sole contribution to the work.

6) None of the Wireless single-lead or 12-lead ECG data have been presented in visual shape to see how the ECG of various normal and abnormal patients look in this specific case.

7) If there is any work available in the literature related to your idea please include a comparitive analysis in the results section and show the superiority of your work both in the table and in the results description sections. 

Author Response

Reviewer 2:

Comment 1: The experimental results presented in fig. 3 are quite confusing; moreover, the figures are of low quality, kindly improve the quality of the figure and text font size so that the audience may understand the data presented.

Thank you for your comment. We have improved to resolution and increased the text font size of the figure to help the readers better understand the results.

Comment 2: why did the authors have chosen traditional statistical analysis methods such as mean and standard deviations have been considering for analysis instead of using modern machine learning, regression, and classification methods?

Reply: We did consider doing more advanced analyses on our data. However, since this is the first study exploring the agreement between 12-lead and single-lead ECGs for ST-segment deviations, we decided that it was more appropriate to present the results in an easy to interpret fashion. This was also to highlight the interesting finding: That single-lead and 12-lead ECGs are comparable in regard to detecting ‘actual’ cardiac stress.

Comment 3: The conclusion section missed the final analysis and factual/tabular conclusion statements about the overall analysis

Reply: We have rewritten the conclusion so that the final analysis is addressed

Comment 4: The authors have not explored the related works in the literature with in the manuscript. We suggest reading related works and logically presenting them in the manuscript in chronological order.

Reply: Thank you we have added articles to the discussion section as line number 238-266 that support the technological foundation for our work but are not aware of other manuscripts with a methodology similar to ours.

Comment 5: What is the actual author's contribution to the work? The author has not even clearly described anywhere their sole contribution to the work.

Reply: thank you. We have added a section regarding the contributions at after the conclusion.

Comment 6: None of the Wireless single-lead or 12-lead ECG data have been presented in visual shape to see how the ECG of various normal and abnormal patients look in this specific case.

Reply: A visual representation of the ST-segment data on a normal and pathological patien has been added to the manuscript as figure 2A and B, line number 150

Comment 7: If there is any work available in the literature related to your idea please include a comparitive analysis in the results section and show the superiority of your work both in the table and in the results description sections.

Reply: No, we are not aware of other studies with a similar methodology including PET-scans, please se our answer to question four.

Round 2

Reviewer 2 Report

The author has tried to address my comments by improving figure, litterature, conclusions etc. i will accept the manuscript in the present form.

Author Response

Thank you for reviewing our article and accepting it in the present form.